# Open Rule Induction

**Wanyun Cui**,* **Xingran Chen**
Shanghai University of Finance and Economics
cui.wanyun@sufe.edu.cn, xingran.chen.sufe@gmail.com

## Abstract

Rules have a number of desirable properties. It is easy to understand, infer new knowledge, and communicate with other inference systems. One weakness of the previous rule induction systems is that they only find rules within a knowledge base (KB) and therefore cannot generalize to more open and complex real-world rules. Recently, the language model (LM)-based rule generation are proposed to enhance the expressive power of the rules. In this paper, we revisit the differences between KB-based rule induction and LM-based rule generation. We argue that, while KB-based methods inducted rules by discovering data commonalities, the current LM-based methods are "learning rules from rules". This limits these methods to only produce "canned" rules whose patterns are constrained by the annotated rules, while discarding the rich expressive power of LMs for free text.

Therefore, in this paper, we propose the open rule induction problem, which aims to induce open rules utilizing the knowledge in LMs. Besides, we propose the Orion (open rule induction) system to automatically mine open rules from LMs without supervision of annotated rules. We conducted extensive experiments to verify the quality and quantity of the inducted open rules. Surprisingly, when applying the open rules in downstream tasks (i.e. relation extraction), these automatically inducted rules even outperformed the manually annotated rules. [2]

## 1 Introduction

Rules induction is a classical problem aiming to find rules from datasets [5, 7]. Previous work has focused on discovering rules within a system. For example, one of the core tasks of inductive logic programming (ILP) is to mine shared rules in the form of Horn clauses from data. Early studies are mainly applied to relatively small relational datasets. Since the axioms of rules is limited to the existing entities and relations within the datasets, the expressiveness of such rules is limited and brittle. John McCarthy pointed out these rules lack commonsense and *"are difficult to extend beyond the scope originally contemplated by their designers"* [18]. In recent years, with the emergence of large-scale knowledge bases (KB) [3] and open information extraction [4] systems, rules mined from them (e.g. AMIE+ [8], Sherlock [26]) are built on a richer set of entities and relations. Nevertheless, both the quantity of knowledge and the complexity of rules are still far weaker than the real-world due to the expressive power of these knowledge bases.

Recently, with the rapid development of pre-trained language models (LM) [6, 22], researchers have found that pre-trained LMs can be used as high-quality open knowledge bases [21, 28] and commonsense knowledge bases [27, 25]. Based on the expression of natural language for complex relationships, the LM as a knowledge base can be used to generalize rules with more expressive power. Based on LMs, Comet [14] proposed to generate new rules (if-then clauses) for arbitrary texts. The generative model is trained on annotated if-then rules in the form of natural language.

---

*Corresponding author
[2]Code and datasets are available at https://github.com/chenxran/Orion

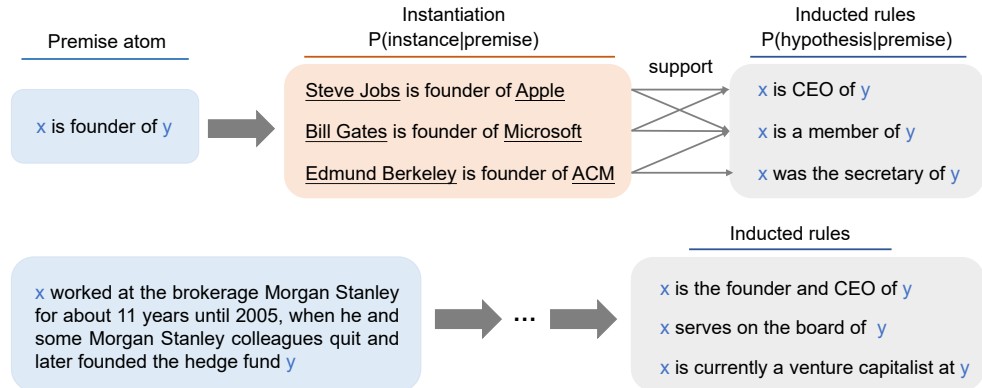

Figure 1: Inducting rules from LMs. We show running examples of Orion.

Therefore, Comet relaxes the form of knowledge in rules from a structured KB, to open natural language. We know that Comet focuses on some specific domains (e.g. social interactions), but when we want to use Comet for open rule generation, training based on annotated rules will constrains the generated rules. Since there are only 23 different types of relations in Comet's training corpus (i.e. ATOMIC2020 [14]), the rules inducted by Comet are limited to these types (e.g. *As a result, PersonX feels*). Besides, the model only learns patterns from the manually annotated rules, which restricts its ability to generate novel rules.

Therefore, we revisit the differences between KB-based rule induction and LM-based rule generation methods. We argue that, leaving aside the difference in knowledge forms, their main difference is that the KB-based method inducts rules by *observing common pattern of a group of entities from the data*, while the LM-based method only *learns from the annotated rules*. Namely, **the former inducts rules from data, while the latter learns rules from rules.** In this case, although the LM contains far more open knowledge than the KB, the current LM-based method still only generates "canned" rules. We believe that the current LM-based rule generation method departs from the principle of KB-based rule induction, i. e., to summarize the commonality of data.

In this paper, we propose to recapture the commonality of the data in the LM-based method, i.e., using the LM itself as a knowledge source and discover commonalities within it. We first proposed the open rule induction problem based on the commonality of knowledge in LMs. Then, we proposed an unsupervised open rule induction system, Orion. Instead of training the LM to generate rules that conform to human annotations, we propose to let the LM "speak" the commonalities of the data by itself. In this way, rules are not constrained by annotations.

To capture data patterns and mine rules directly from LMs, we use prompts to probe the knowledge of LMs [21]. Prompts' predictions reflect the knowledge in LMs. For example, in Fig. 1, based on prompt: *x is founder of y*, we probe instances such as Steve Jobs-Apple, Bill Gates-Microsoft, etc. From these instances, we further use LM to induct other expressions they support, such as *x is CEO of y*, etc. Notice that unlike Comet which learns from annotated rules, we directly mine the patterns from knowledge in LMs. In addition, taking full advantage of the expressive power of LMs, we inducted rules for the complex example at the bottom of Fig. 1.

The significance of our proposed open rule induction in this paper is twofold. First, from the perspective of rule induction, we gain more expressive rules to approach the real-world. And by adding these rules to downstream tasks, the effect can be significantly improved. We will empirically verify this in Sec 6.1. Second, from the LM perspective, rules can identify potential errors in language models. For example, in Fig. 1, if we wrongly inducted the hypothesis *x was the secretary of y*, this suggests that the LM has some kind of cognitive bias. We will elaborate this in Sec 6.2.

## 2 Open Rule Induction Problem

### 2.1 Preliminary: Rules in KB-based Methods

We refer to the definition of rules based on the Horn clause in KB-based rule induction to help define our problem. In a Horn clause, an atom is a fact that can have variables at the subject and/or object

position. For example, $(x, founderOf, y)$ is an atom whose subject and object are variables. A Horn clause contains a head and a body, where the head is an atom, and the body is a collection of atoms. A rule is then an implication from body to head: $body_1 \wedge \cdots body_n \Rightarrow head$. An *instantiation* of a rule is a copy of that rule, where all variables are replaced by specific instances in the KB.

## 2.2 Problem Definition

We define the atom of an open rule as a natural language sentence describing some relationship between subject and object. For simplicity, we assume both subject and object are variables. For example, *(x, is founder of, y)* is an atom which describes the relation between $x$ and $y$. In defining the open rule, for simplicity, we only consider the bodies with one atom. As we will describe in Sec 3, the open rule can be easily extended to more variables by slightly modifying prompts. We define an open rule as a derivation from a premise atom to a hypothesis atom.

**Definition 1** (Open rule). *An open rule is a implication from the premise atom $(x, r_p, y)$ to hypothesis atom $(x, r_h, y)$:*

$$(x, r_p, y) \Rightarrow (x, r_h, y) \tag{1}$$

*where $r_p$ and $r_h$ are natural language descriptions. The rule implies that instance pairs with $r_p$ relation also (very often) has $r_h$ relation.*

For example, the open rule $(x, \text{is founder of}, y) \Rightarrow (x, \text{is CEO of}, y)$ means that (very often) the founder of an organization is the CEO of it. And *(Steve Jobs, is founder of, Apple) ⇒ (Steve Jobs, is CEO of, Apple)* in an instantiation of the rule. As we will show in Sec 3, generalizing the open rules to more than two variables is easy. And we will discuss the potential extension of the open rules with new variables in Appendix **??**.

To model to the uncertainty of the open rule, in our problem definition, we use probability to represent the open rule: $P_{LM}(r_h|r_p)$ denotes the probability of inferring hypothesis $(x, r_h, y)$ from premise $(x, r_p, y)$. $P_{LM}()$ means the probability is derived from the language model $LM$. For simplicity, we will use $P()$ instead of $P_{LM}()$ in the rest of this paper.

For a given premise atom $(x, r_p, y)$, we want to induct $k$ most relevant hypothesis atoms from the LM. Specifically, we define the problem as follows:

**Problem Definition 1** (Open rule induction). *For a given premise atom $(x, r_p, y)$ and $k$, find top $k$ $r_h$ w.r.t. $P(r_h|r_p)$.*

We compute $P(r_h|r_p)$ by the marginal distribution of the instantiation ($ins$) as below:

$$P(r_h|r_p) = \sum_{ins} P(r_h|ins, r_p)P(ins|r_p) \tag{2}$$

where $P(ins{=}(x_0, y_0)|r_p)$ denotes the conditional probability distribution of $x_0, y_0$ corresponding $r_p$. And $P(r_h|ins{=}(x_0, y_0), r_p)$ denotes the conditional probability of $(x_0, r_h, y_0)$ given $(x_0, r_p, y_0)$ and the specific instance $(x, y)$. For example, $P(ins{=}(\text{Steve Jobs,Apple})|r_p{=}\text{is founder of})$ denotes the probability of instance $x{=}$Steve Jobs and $y{=}$Apple for the relation *is founder of* in the LM. And $P(r_h{=}\text{is ceo of}|ins{=}(\text{Steve Jobs,Apple}), r_p{=}\text{is founder of})$ denotes the conditional probability of $(\text{Steve Jobs, is ceo of, Apple})$, given that $(\text{Steve Jobs, is founder of, Apple})$.

Note that, when $ins{=}(\text{Steve Jobs,Apple})$ is known, whether the instantiation has relation *is ceo of* is also known. That is, given the $ins$, $r_h$ is independent from $r_h$. So we have:

$$P(r_h|r_p) = \sum_{ins} \underbrace{P(r_h|ins)}_{\text{Applicability}} \underbrace{P(ins|r_p)}_{\text{Instantiation}} \tag{3}$$

## 2.3 Rationale of the Open Rule Induction Problem

We use the concept of *support* in KB-based rule induction to explain Eq. (3) and Problem 1. Support is the core metric in KB-based rule induction, which quantifies the amount of instances following the rule in the dataset. We will analogize the terms in Eq. (3) to the factors of support as below.

**Instantiation** First, we consider $P(ins|r_p)$ as the instantiation by the language model. The probability can be considered as the *typicality* of this instance for $r_p$ by the LM.

**Applicability** Second, we consider $P(r_h|ins)$ in Eq. (3) as the applicability of $r_h$ to $ins$ in the LM. For example, for a well-trained LM, we can get $P(r_h = $ is ceo of|Steve Jobs, Apple) with a high probability.

In summary, Eq. (3) denotes the expected number of instantiations that can be applied to the hypothesis atom w.r.t. their applicability. This is consistent with the idea of support in the KB-based rule induction. We will elaborate how to compute $P(ins|r_p)$ and $P(r_h|ins)$ via the LM in Sec 3.

## 3  Masked Language Models for Relational Descriptions

In this section, we illustrate how to calculate the two terms in Eq. (3), i.e., $P(r_h|ins)$ and $P(ins|r_p)$. For LMs, both probabilities can be considered as special cases of the masked language model (MLM), i.e. $P(mask|masked\ sentence)$. From the MLM perspective, $P(r_h|ins = (x,y))$ is equivalent to predicting the masked text between $x$ and $y$, while $P(ins = (x,y)|r_p)$ is equivalent to predicting the masked $x, y$ for given $r_p$. Since MLM is a typical goal of the language model pre-training, we can directly use existing pre-trained LMs to predict the masks.

**Probability probing via prompts** To compute the two probabilities, we construct following prompts:

- For $P(r_h|ins = (x,y))$: x <mask> y.
- For $P(ins = (x,y)|r_p)$: <mask>$_x$ $r_p$ <mask>$_y$.

For example, we use the prompt *Steve Jobs <mask> Apple.* to compute $P(r_h|ins = $ (Steve Jobs,Apple)), and the prompt *<mask>$_x$ is founder of <mask>$_y$.* to compute $P(ins = (x,y)|r_p = $ is founder of). In our problem, one mask may correspond to multiple tokens. Therefore we use Bart [16] as our LM, which uses seq2seq to decode one or more tokens for each mask.

Although we only consider rules with two variables in this paper, our method can be easily generalized to arbitrary number of instances by modifying the prompts (e.g. from *x <mask> y.* to *x <mask> y <mask> z*).

**Relational description generation via weak supervision** We noticed that MLM are not directly applicable to computing $P(r_h|ins)$ and $P(ins|r_p)$. For $P(r_h|instances)$ we require the LM to generate text describing exactly the relationships between $x$ and $y$. Similarly, for $P(instances|r_p)$, the generated text fragments are required to be exactly instances/entities. However, the original MLM task for language model pre-training does not have these restrictions. It may predict arbitrary descriptions which leads to a lot of noise. For example. Bart predicts "Tokyo Metropolitan Government,Japan,Japan.Tokyo, Japan." for prompt *Tokyo <mask> Japan.*

To ensure that the LM correctly predicts $P(r_h|ins)$ and $P(ins|r_p)$, we continue training the LM on relational description corpora. We obtain a large-scale relational description corpus using the weak supervision technique [15], and use it to construct two separate training corpora for $P(r_h|ins)$ and $P(ins|r_p)$, respectively. For $P(r_h|ins)$, we only mask out text except the named entities, and train the LM to predict the mask. For $P(ins|r_p)$, we only mask out named entities.

Specifically, starting from the Wikipedia and Bookcorpus [32], we searched for sentences containing one or two named entities. We consider that these sentences are (likely) describing the relationship between entities. We continue to train two Bart models on each of these two corpora, which are used to predict $P(r_h|ins)$ and $P(ins|r_p)$, respectively. We used the Spacy NER library to automatically extract the named entities. We filtered the entities related to dates and numbers. The resulting dataset consists of 93.63 millions samples. The continuing train plays the role of denoising the generative process in Eq. (3), which is also used in [24].

## 4  Supported Beam Search for Rule Decoding

Another challenge of solving Problem 1 is to efficiently generate $r_h$ of *a crowd of instantiations*. Although we can compute $P(r_h|ins)P(ins|r_p)$ according to Sec 3, computing Eq. (3) is still intractable: (a) the search space grows exponentially according to the length of the generated rule; (b) we need to find the $r_h$ that has the top $k$ probability among *all* instantiations, rather than for a single instantiation as in standard decoder.

**Beam search for instantiation** First, for the exponential number of all possible instances, we use the model for $P(ins|r_p)$ trained in Sec 3 to generate the top $k$ instantiations, denoted as $INS$:

$$INS = topk_{ins}(P(ins|r_p)) \tag{4}$$

We use *beam search*, a common and efficient way for decoding $INS$. Beam search is a search heuristic to maintain a beam of size $k$ containing a set of probable outputs. It generates $r_h$ from beginning to end, conditioning on the instances and already-generated text.

We use these $k$ instances to approximate Eq. (3).

$$P(r_h|r_p) \approx \sum_{ins \in INS} P(r_h|ins)P(ins|r_p) \tag{5}$$

**Supported beam search** To solve Problem 1, we need to consider the support from different instantiations. A straightforward method is to first generate top $k$ $r_h$ for $ins \in INS$ separately with beam search, and then fuse these $r_h$. However, this risks making locally optimal decisions which are actually globally sub-optimal for all instantiations. These top $k$ beams for each individual instantiation may not be shared by different instantiations. Therefore, in order to decode the rules, we need to consider the support of all instantiations in our decoding heuristic.

We propose supported beam search (STS) which decodes the open rules by considering all instantiations. The probability of generating the next word considering all instantiations is:

$$
\begin{aligned}
P(beam' = beam + w|r_p) &= P(w|beam, r_p)P(beam|r_p) \\
&\approx \sum_{ins \in INS} P(w|r_p, beam, ins)P(ins|beam, r_p)P(beam|r_p) \quad \text{(Eq. (5))} \\
&= \sum_{ins \in INS} P(w|r_p, beam, ins)P(beam|ins, r_p)P(ins|r_p) \quad \text{(Bayesian rule)} \\
&= \sum_{ins \in INS} P(w|beam, ins)P(beam|ins)P(ins|r_p) \quad \text{(Independence)}
\end{aligned} \tag{6}
$$

where $beam' = beam + w$ means appending the word $w$ at the end of $beam$, which forms a longer beam. $P(w|beam, ins)$ can be computed via fine-tuned Bart of $P(r_h|ins)$ in Sec 3. $P(beam|ins)$ is computed and updated by:

$$P(beam' = beam + w|ins) = P(w|ins, beam)P(beam|ins) \tag{7}$$

Here $P(beam' = beam + w|r_p)$ can be considered as the global score of $w$, as it aggregates different instantiations. And $P(beam' = beam + w|ins)$ can be considered as the local score of $w$, as it only considers the instantiation of $ins$.

**Algorithm** In our implementation, we decode different instantiations in $INS$ simultaneously via batches. We assemble different instances of $ins \in INS$ into one batch. In each step of the decoding, we aggregate the local scores of $w$ to compute its global score. We uniformly select the top $k$ words w.r.t. their global scores for all instantiations. That is, we maintain identical beams for different instances in the batch, instead of maintaining individual beams for each instance individually.

---

**Algorithm 1:** Supported beam search

1 **Function** SupportedBeamSearch($r_p$, $INS$, $k$):
2     $batch\_beams \leftarrow \{NULL\}$
3     $P(beam = NULL|ins) \leftarrow 1$ for $ins \in INS$
4     $P(beam = NULL|r_p) \leftarrow 1$
5     **for** *timestep* $t = 1 \cdots T$ **do**
6         Update $P(beam'|ins)$ for $len(beam') = t$ and $ins \in INS$ by Eq. (7)
7         Update $P(beam'|r_p)$ for $len(beam') = t$ by Eq. (6)
8         $batch\_beams \leftarrow topk_{beam'}(P(beam'|r_p))$     // Greedily select top $k$ beams w.r.t. their global scores.
9     **return** $batch\_beams$

---

We show the pseudo-code of supported beam search in Algo. 1. Unlike the traditional beam search which maintains separate beams for different samples in a batch, we maintain a set of shared beams for all instances, denoted as $batch\_beams$. At each timestamp, we update the local score $P(beam|r_p, ins)$ and global score $P(beam|r_p)$ in turn. Then in line 8 we greedily selects the top $k$ beams w.r.t. the $P(beam|r_p)$, i.e., the approximated goal of Problem 1.

# 5 Experiments

All the experiments run over a cloud of servers. Each server has 4 Nvdia Tesla V100 GPUs.

## 5.1 Datasets

**Manual constructed dataset** To evaluate the effectiveness of open rule induction, we constructed our own benchmark dataset with 155 premise atoms. We call it "OpenRule155" for convenience in this section. First, to construct premises describing different relationships between $x$ and $y$, we collect 121 premise relations from 6 relationship extraction datasets (Google-RE [21], TREx [21], NYT10 [23], WIKI80 [9], FewRel [9], SemEval [11]) and one knowledge graph (Yago2). We converted all relations of these datasets into premise atoms, and obtained a total of 121 premise atoms after removing duplicates. We also selected 34 relations from Yago2. We select these 34 relations because they occur frequently in the bodies of inducted rules by AMIE+ and RuLES. So we think these relations have a higher inductive value. We asked the annotators to annotate each premise with 5 hypothesis. We filtered out duplicates and low quality hypothesis and ended up with an average of 3.3 hypothesis atoms for each premise.

**Converting relations to premise atoms** We convert a relation into a premise atom via the template *(x, is relation of, y)* or *(x, relation, y)*. For example, the relation *<founderOf>* and *<believeIn>* in Yago2 will be transformed into *(x, is founder of, y)* and *(x, believe in, y)*, respectively.

**Relation extraction datasets** We also conducted experiments over relation extraction datasets to evaluate whether Orion extracts the annotated relations from given texts. We use Google-RE, TREx, NYT10, WIKI80, FewRel, SemEval as the relation extraction datasets.

## 5.2 Baselines

**LM-based baselines** We use the following LM-based rule generation baselines:

1. **Comet:** The input of Comet is a premise atom, and the output is a collection of hypothesis atoms with different relations. To compare with the top 10 rules inducted by Orion, we select 10 relations of Comet. See Appendix for the list of relations. Note that the hypothesis atoms generated by Comet are not always describing the relationship between $x$ and $y$, but may be describing about $x$ only. For each selected relation of Comet, we generate 10 hypothesis atoms. If there are hypothesis atoms of both $x$ and $y$, we choose the one with the highest probability. Otherwise, we choose the one with the highest probability among the descriptions of $x$.

2. **Prompt:** Inspired by the work on prompt-based knowledge extraction from LMs [21], we proposed a prompt-based method as a baseline. We use the prompt: *if $r_p$ then <mask>.* and take the LM's prediction for <mask> as $r_h$. Specifically, we use Bart as the LM and select the top 10 predictions as $r_h$.

3. **Prompt (fine-tuned):** We came up with a stronger baseline by fine-tuning the above prompt model. We collect a set of such sentences "if sent1 then sent2" from Wikipedia and Bookcorpus and mask sent2. Then we fine-tune the prompt model over these sentences.

**KB-based baselines** We use AMIE+ [8] and RuLES [12] as the KB-based rule induction baselines.

**Ablations** We also considered the following ablation models.

- **Without continuing training** $P(r_h|ins)$ **or** $P(ins|r_p)$ In Sec 3, we propose to generate relational descriptions by continuing training Bart. To verify its effect, we replace the models for $P(r_h|ins)$ or $P(ins|r_p)$ with the original Bart as an ablation.

- **Without STS** To verify the effectiveness STS, we replace STS with the original beam search. When decoding, we first use beam search for each $ins \in INS$ separately to generate the top $k$ $r_h$ w.r.t. $P(r_h|r_p, ins)$. Then we aggregate these hypothesis atoms according to Eq. (5) and select the top $k$ of them.

## 5.3 Main Results

Table 1: Results over the OpenRule155.

| Our Dataset | BLEU-1 | BLEU-2 | BLEU-4 | ROUGE-L | METEOR | self-BLEU-2 |
|---|---|---|---|---|---|---|
| Prompt | 17.77 | 3.65 | 0.48 | 18.65 | 12.94 | 86.63 |
| Prompt (fine-tuned) | 20.95 | 7.58 | 0.86 | 22.37 | 17.24 | 82.13 |
| Comet | 21.58 | 8.15 | 1.04 | 23.45 | 5.44 | 90.78 |
| Orion - STS | 44.92 | 20.24 | 1.21 | 49.72 | 39.68 | 89.84 |
| Orion - train $P(ins|r_p)$ | 15.85 | 3.11 | 0.00 | 32.91 | 13.19 | 90.29 |
| Orion - train $P(r_h|ins)$ | 19.17 | 3.05 | 0.07 | 34.99 | 10.30 | **83.54** |
| Orion | **45.41** | **21.29** | **1.30** | **50.37** | **40.41** | 90.94 |

We report the performance of the models on the OpenRule155 in Table 1. We use BLEU-1/2/4 [20], ROUGE-L [17], and METEOR [2] to evaluate whether the model-inducted $r_h$ is similar to the manually annotated hypothesis. We also report the self-BLEU-2 [31] of the model, which is used to measure the diversity (the smaller the more diverse).

We compare Orion with the LM-based baselines. From the perspective of quality, Orion significantly outperforms the baselines. From the perspective of generated diversity, the diversity of Orion is also competitive with Comet.

**Ablations** We also compare with the ablation models in Table 1. First, we find that the effectiveness of the models reduces after removing any module. This verifies the effectiveness of the proposed modules in this paper. In particular, we find that continuing train $P(r_h|ins)$ has the most significant effect on the model. As we mentioned in Sec 3, this is because that without continuing training, Bart easily produces noisy text that does not describe the relationship between $x$ and $y$.

## 5.4 Comparison with the KB-based Rule Induction

In Sec 1, we claimed that open rules are more flexible than rules inducted from KBs. In this subsection, we verify this by comparing the results of Orion and KB-based rule induction systems.

**Rules from KB-based induction** Specifically, we compared the rules inducted by Orion and by the KB-based methods AMIE+ [8] and RuLES [12]. We use AMIE+ and RuLES to induct rules on Yago2 [13]. For a fair comparison, we also only retain the rules generalized by AMIE+ and RuLES which contain exactly two variables. AMIE+ and RuLES mined 115 and 47 rules that meet the requirement, respectively.

**Comparing open rule induction with KB-based Horn rules** In order to verify the inductive ability of Orion, for each Horn rule $body \Rightarrow head$ inducted by the KB-based methods, we converted the relation of $body$ into a premise atom according to the conversion method in Sec 5.1. AMIE+ and RuLES have 24, 20 different bodies, respectively. For each converted premise atom, we use Orion to induct $k = 5, 10, 20$ corresponding open rules. We manually evaluate whether these inducted rules are correct. During the human evaluation, we require a correct rule to be plausible, unambiguous, and fluent. For example, we label "[X] is a provincial capital of [Y] $\Rightarrow$ [X] is the largest city in [Y]" as correct, because this inference is plausible to be valid. In contrast, we will label "[X] lives in [Y] $\Rightarrow$ [X] grew up in the east end of [Y]" as incorrect, because the probability that [X] happens to grow up on the east end is too low.

The results are shown in Table 2. It can be found that the accuracy of Orion is competitive with AMIE+ and RuLES. As $k$ increases, Orion keeps finding new rules without decrease in accuracy. Note that besides the bodies covered by AMIE+ and RuLES, Orion also generalizes rules from novel premises. This indicates that Orion finds substantially more rules than the KB-based methods with competitive quality.

Table 2: Comparisons with KB-based methods.

| | Accuracy | #Rules |
|---|---|---|
| AMIE+ | 55.7 | 115 |
| RuLES | 51.1 | 47 |
| Orion (k=5) | 50.0 | 120 |
| Orion (k=10) | 49.2 | 240 |
| Orion (k=20) | 51.3 | 480 |

## 5.5 Effect for Complex Premises

Orion is able to generate rules for complex premises, as the pre-training corpora of LMs contain extensive complex texts. We show two examples from TREx and FewRel in Table 3 with ($k = 5$). It can be seen that Orion generates valid rules for complex and long texts. On the other hand, the rules generated by Comet are often only about $x$, not about the relationship between $x$ and $y$. And these rules are often about human characteristics, even if $x$ is a country in case 1. This is due to the limitation of Comet's training data that leads to the bias of the generated rules.

Table 3: Effect of complex rule induction. Original sentence of **Case 1**: *[X]'s emergence from international isolation has been marked through improved and expanded relations with other nations such as [Y], France, Japan, Sweden, and India.* **Case 2**: *His guitar work on the title track is credited as what first drew [X] to him, who two years later invited allman to join him as part of [Y].*

| | Orion | Comet |
|---|---|---|
| **C1** | [X] has a long history of diplomatic relations with [Y]. 
 [X] is the largest exporter of oil to [Y]. 
 [X]'s economy is heavily dependent on [Y]. 
 [X]'s foreign policy is based on its close relationship with [Y]. 
 [X] has been the largest exporter of uranium to [Y]. | <xReact>: happy. 
 <xReason>: [X] is no longer isolated. 
 <xWant>: to make new friends. 
 <isAfter>: [X] gets a new job. 
 <isBefore>: [X] has a better relationship with [Y]. |
| **C2** | [X], guitarist and singer of [Y]. 
 [X] and his band [Y]. 
 [X] has been a fan of [Y]. 
 [X] was a fan of [Y]. 
 [X] was a fan of the band [Y]. | <xReact>: happy. 
 <xReason>: his guitar work. 
 <xWant>: to play a song. 
 <isAfter>: [X] plays guitar on the song. 
 <isBefore>: his guitar work. |

## 6 Application

### 6.1 Introducing Open Rules in Relation Extraction

**Setup** We apply the inducted open rules to relation extraction to verify its value. To do this, We introduce the inducted open rules as extra knowledge, and evaluate whether the inducted rules improve the effect of relation extraction models. We used ExpBERT [19] as the backbone. ExpBERT introduces several textual descriptions of all candidate relations as external knowledge. For example, for the relation *spouse*, ExpBERT introduces external knowledge in the form of *x's husband is y* into the BERT model. The original descriptions of each relation in ExpBERT comes from manual annotation. To verify the effect of inducted open rules, we replace these manual annotated rules with the open rules inducted by Orion.

Specifically, We follow the settings in ExpBERT and use the Spouse and Disease [10] for evaluation. For each relation, we construct the corresponding premise atom according to Sec 5.1. We following the settings of ExpBERT and use $k = 29, 41$ hypothesis atoms inducted by Orion as for Disease and Spouse, respectively. For example, for relation *spouse*, Orion generates hypothesis atoms like *(x, is married to, y)* as the external descriptions. In addition, we modified ExpBERT to allow the training process to fine-tune the parameters that were frozen in the original ExpBERT, as we found that this will improve the model's effectiveness.

**Results** From Table 4, our automatic inducted rules even outperforms the manually annotated rules. We think that this is because Orion's rules are more unbiased and diverse than the manual annotations. This strongly verified the applicability of the open rules.

Table 4: F1 scores on relation extraction tasks. The *annotated rules* are from the original ExpBERT. Averaged over 5 runs.

|  | Spouse | Disease |
|---|---|---|
| BERT | $46.43 \pm 0.84$ | $40.20 \pm 2.43$ |
| ExpBERT + annotated rules | $76.04 \pm 0.47$ | $56.92 \pm 0.82$ |
| ExpBERT + inducted open rules | $\mathbf{76.05 \pm 0.52}$ | $\mathbf{57.68 \pm 1.34}$ |

**Coverage evaluation** We also directly evaluate whether the open rules inducted by Orion cover the target relation. For the bottom example in Fig. 1, since we can extract the relation *founder* from *x worked ... founded the hedge fund y.*, we expect the model to also induct *x is founder of y* given the premise.

For this purpose, we transform the samples in the relation extraction dataset into premise atoms by replacing the entity pairs with $x$ and $y$, respectively. We induct rules from these premise atoms and evaluate whether the hypothesis atoms cover the corresponding relations. For each target relation, we convert it to a hypothesis atom via the method in Sec 5.1, and use the converted hypothesis atom as the ground truth. We compute the correlation between the inducted $r_h$ and the ground truth.

We report the results of FewRel, NYT10, WIKI80, TREx, Google-RE, and SemEval in Table 5. Since the number of samples is different for different datasets and relations, in order to get a uniform evaluation, we select 5 training samples uniformly as premise atoms for each relation in each dataset. Note that there are $k = 10$ hypothesis atoms for each premise. To evaluate the coverage, we report the one with the highest score. Orion outperforms the baseline by a large margin.

Table 5: Results on relation extraction datasets.

|  | FewRel | NYT10 | WIKI80 | TREx | Google-RE | SemEval |
|---|---|---|---|---|---|---|
| | | | BLEU-2/4 | | | |
| Comet | 0.60/0.00 | 0.73/0.00 | 0.36/0.00 | 0.82/0.11 | 0.60/0.00 | 1.80/0.00 |
| Prompt | 0.95/0.00 | 0.64/0.00 | 0.87/0.00 | 1.45/0.11 | 0.79/0.00 | 0.90/0.00 |
| Orion | **9.08/0.08** | **8.34/0.51** | **7.24/0.30** | **9.03/1.14** | **9.22/0.00** | **5.69/0.00** |
| | | | ROUGE-L | | | |
| Comet | 3.02 | 5.70 | 2.44 | 4.77 | 4.19 | 7.82 |
| Prompt | 15.35 | 15.10 | 16.38 | 15.80 | 16.44 | 12.54 |
| Orion | **38.72** | **36.72** | **36.75** | **39.51** | **36.93** | **37.90** |
| | | | METEOR | | | |
| Comet | 1.60 | 3.03 | 1.19 | 2.14 | 2.08 | 4.60 |
| Prompt | 9.94 | 9.84 | 10.87 | 11.50 | 8.41 | 8.47 |
| Orion | **25.28** | **25.78** | **23.8** | **26.29** | **24.67** | **26.71** |

### 6.2 Application: Error Identification in Language Models

We use inducted rules to identify potential errors in the pre-trained LM. Some rules that defy human commonsense are incorrectly inducted. This is actually due to the bias of the language model. Specifically, we found the bias of the language model caused by its pre-training corpus. We list some examples in Table 6.

## 7 Related Work

As a classical problem, rule induction has gained extensive studies. Related researches include mining association rules [1], logical rules [29], etc. Traditional methods of rule induction tend to work only on small knowledge bases. In recent years, with the emergence of large-scale open knowledge graphs [13] and open information extraction systems [4], studies have focused on mining rules from such large-scale knowledge bases [26, 8]. However, the representation capability of even

Table 6: Examples of identified errors

| |
|---|
| **Inducted rule**: [X] is the politician of [Y]. ⇒ [X] was the founder and president of [Y]. 
 **Identified error**: The training corpus description of politician has a disproportionate number of founder and president entities. This led to a bias in LM's perception: it assumes that politician is always founder and president. |
| **Inducted rule**: [X] is an instance of [Y]. ⇒ [X] is a lower house of [Y]. 
 **Identified error**: The frequency of political entities in the training corpus is too high. The LM tends to generate political entity descriptions. |

the largest knowledge bases still do not approximate the real-world, especially from the perspective of commonsense.

On the other hand, reserchers found that pre-trained LMs can be used as open knowledge bases [21, 28] and commonsense knowledge bases [27, 25]. Due to the unstructured form of knowledge representation, the language model is much more capable for representing open knowledge. Using the language model as a basis, we want to mine open rules. Comet [14] is a relevant attempt. It learns from ATOMIC2020 [14] to generate rules in the form of if-then clauses. However, influenced by the manual annotation of ATOMIC2020, Comet's rules are "canned" and repetitive [30]. In this paper, we want to generate rules unsupervised directly from LM's knowledge, thus achieving open rule induction.

## 8 Conclusion

For rule induction, in order to break the limitation of representation in KBs, we propose to use unstructured text as atoms of rules. Based on pre-trained language models, we propose the open rule induction problem. To solve this problem, we propose the Orion system, which extracts rules from the language model completely unsupervised. We also propose to optimize Orion by continuing training the language model, as well as the decoding heuristic.

We conducted a variety of experiments. We verified that the effectiveness of Orion exceeds that of LM-based and KB-based baselines. In addition, an application to the relation extraction task found that the model with Orion's inductive rules even outperformed that of manually annotated rules. We also used Orion's rules to identify potential errors in language models.

## Acknowledgments and Disclosure of Funding

We thank Wenting Ba for her valuable plotting assistance. This paper was supported by National Natural Science Foundation of China (No. 61906116), by Shanghai Sailing Program (No. 19YF1414700).

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
