# OpenReview forum: "Open Rule Induction"
_NeurIPS.cc/2021/Conference — NeurIPS 2021 Poster_

### Official Review · Reviewer_hJMv · 2021-07-06

**Rating:** 6
**Confidence:** 3

**Summary:**

The paper proposes a system for rule induction that uses unstructured text as rule atoms in order to break the representation limitation in KBs. The system, called Orion, uses pre-trained language models to solve the problem called in the paper "open rule induction" by extracting rules in a completely unsupervised way. The paper also presents a series of experiments showing that Orion outperforms baselines based on LM and KB. The results obtained are promising and, in general, very good.

**Limitations And Societal Impact:**

Limitations have been adequately discussed.

**Main Review:**

The proposal is interesting and looks promising. I did not find major flaws in the definitions of the proposal.

The paper is easy to read and clear. I only have two small comments, which are as follows:
- in Figure 1, it is not clear the transition from what written in the blue box in the second line, and what is in the grey box. This is explained later in the paper. Maybe here, the image could be improved by replacing those "..." with information about the process. The inducted rules are very different from those in the blue box.
- lines 133-135, the paper says that the method can be easily generalised to an arbitrary number of instances. It would be interesting here to read about possible problems and limitations due to this step. This is out of the scope of this paper, but I think it would increase its quality.



On line 124 langauge -> language
In Algorithm 1, in the comment  Geedily -> Greedily

AFTER REBUTTAL
After reading all the reviews and comments, and after a discussion with the other reviewer, I think that on the one hand there are several flaws in the article but, on the other hand, it is acceptable for NeurIPS. To reflect this, I have decided to lower my score but it will remain above the acceptance threshold.

**Time Spent Reviewing:**

5

---

> ### Author Response · Authors · 2021-08-10
> **Response to Reviewer hJMv**
>
> We thank the reviewer for the positive review and constructive advice. Our responses to the questions are as below:
>
> **Q1. Figure 1 can be improved.**
>
> **Response:** Thanks for the comment! We will improve Figure 1 accordingly.
>
> **Q2. It would be interesting here to read about possible problems and limitations about generalizing the number of instances in rules.**
>
> **Response:**  We further discussed how rules could be extended to multiple body atoms, and the problems that might be encountered. Please refer to the general response.

---

> > ### Comment · Reviewer_hJMv · 2021-09-10
> > **Thank you**
> >
> > I would like to thanks the authors for their answers.
> > As reported in my review, I still think the paper is good and acceptable for NeurIPS. However, the other reviewers have pointed out several aspects in the paper that would require to be better discussed in the paper. Therefore, I decided to slightly lower my score.

---

> > > ### Author Response · Authors · 2021-09-10
> > > **Thank you**
> > >
> > > We understand it would be important yet difficult to reach an agreement among reviewers, and we appreicate that your are still supporting the acceptance of our paper. Thank you.

---

### Official Review · Reviewer_mUXe · 2021-07-11

**Rating:** 5
**Confidence:** 4

**Summary:**

The paper introduces the problem of open rule induction—learning rules from implicit knowledge in pretrained language models (LMs). Then it proposes Orion—a system for open rule induction by probing the knowledge in LMs using prompts.  Experimental results show that Orion can learn many valid rules, and the learned rules are useful for relation extraction.

**Limitations And Societal Impact:**

Orion induces rules from large-scale pretrained language models such as Bart. Researchers have studied the potential negative societal impact of large language models ([A], etc.), e.g., they tend to produce biased texts. It may be worth discussing whether such biases make into the rules learned by Orion. The discussion in Sec. 6.2 touches upon this issue from a different angle but doesn't directly address ethical concerns. A more direct discussion might be helpful. Nevertheless, I don't think this is a deal breaker when considering whether to accept this paper.


[A] Bender, Emily M., et al. "On the Dangers of Stochastic Parrots: Can Language Models Be Too Big?🦜." Proceedings of the 2021 ACM Conference on Fairness, Accountability, and Transparency. 2021.

**Main Review:**

## Strengths


+ The problem of open rule induction is novel. Prior work in rule induction usually learn rules from a fixed dataset—either a structured knowledge graph or an unstructured text corpus. Open rule induction learns rules by probing the implicit knowledge in large-scale pre-trained language models (LMs), which is more flexible and covers more rules compared with a single dataset.

+ The paper proposes Orion—a method for open rule induction that learns rules by querying LMs with carefully constructed prompts.

+ In experiments, the authors compare Orion with several baselines and show that Orion can learn valid rules from LMs. Also, the learned rules are shown to be useful in downstream tasks such as relation extraction.


## Weaknesses and Questions



- The paper made the false claim that the learned rules lead to *significant* improvements on relation extraction (Line 63). In table 4, the performance gap between "ExpBERT + annotated rules" and "ExpBERT + inducted open rules" is small—even smaller than the error bars. It's not clear if the difference is statistically significant.

- The rules in this paper have a restricted form: horn clauses with only one atom in the body. They cover only a small fraction of what can be achieved through rule-based reasoning. Although the paper discussed how the syntactic constraint could be relaxed, it is not clear if the proposed solution will actually work.

- I'm not completely convinced by the main results in Sec. 5.3. Orion and the two baselines are all evaluated on a small-scale dataset constructed in this paper, but they are trained on different tasks and different datasets. I'm not sure if the final results are comparable. Also, it's hard to come up with a reasonable evaluation metric. The paper used BLEU but didn't justify it.

- In Sec. 5.4, the authors manually evaluated the accuracy of the learned rules, which may be subjective and noisy. The accuracies of different methods are close in Table 2. Therefore, it is not clear if the result in Table 2 tells us anything.



**Time Spent Reviewing:**

3

---

> ### Author Response · Authors · 2021-08-10
> **Response to Reviewer muXe**
>
> We thank the reviewer for the valuable feedback. Our responses to the questions are as below:
>
> **Q1. The paper made the false claim that the learned rules lead to significant improvements over ExpBERT on relation extraction (line 63).**
>
> **Response:** We would like to clarify that what we are referring to here is that compared to BERT, the effect could be significantly improved. This comparison is reasonable because our method does not introduce any new manual annotation compared to BERT.
>
> **Q2. The rules in this paper only have one atom in the body.**
>
> **Response:** Our rules could be extended to the case of multiple bodies, although this is not the focus of this paper. Please refer to the general response for our solutions and the problems that might be encountered.
>
> **Q3. It’s hard to come up with a reasonable evaluation metric. The paper used BLEU but didn’t justify it.**
>
> **Response:** Indeed, it’s hard to come up with a reasonable automatic evaluation metric. The metric BLEU used in this paper is from Comet's paper. We have done the following to overcome the difficulties of the evaluation.
> 1. To avoid subjectivity in the evaluation of rules, we demonstrate their effectiveness by applying these rules to a downstream task with objective metrics (accuracy for relation extraction in Sec 6.1).
> 2. Manual evaluation. We require a correct rule to be plausible, unambiguous, and fluent. We show the results of our manual evaluation in Table 2. We also give adequate examples and their evaluation results in the appendix.
>
> **Q4. It is not clear if the result in Table 2 tells us anything.**
>
> **Response:** Table 2 shows that the rules inducted by Orion and the KB-based methods are of competitive quality. But since Orion can also generalize rules from premises not covered by the KB-based methods, it finds substantially more rules than the KB-based methods with high quality. (line 261)
>
> **Societal Impact: The discussion in Sec. 6.2 touches upon the potential negative societal impact from a different angle but doesn't directly address ethical concerns.**
>
> **Response:**
>
> Indeed, the bias of the language model leads to potential bias and ethical issues in rules. This is a limitation of our current work. We believe that this needs to be addressed by training more unbiased LMs.
>
> However, from another perspective, since reducing the bias of LMs is a more fundamental ethical issue, although it is not yet possible to reduce the bias of LMs by our generated rules, these rules help us to discover and evaluate the inferential bias of LMs.

---

### Official Review · Reviewer_cSoa · 2021-07-16

**Rating:** 5
**Confidence:** 4

**Summary:**

This paper considers to induce open rules (r_p -> r_h) from unstructured corpus without any knowledge bases. Its key idea is to model the probability P(r_h | r_p) by introducing a latent variable ins. Then it handles two factorized probabilities P(r_h | ins) and P(ins | r_p) through a masked language model. To further improve the performance, it collects a dataset with labeled entities and performs continue training for the masked language model. Experiments show that the proposed method can induce descent rules to some extent and the induced rules can further be benifical to downstream tasks.


**Limitations And Societal Impact:**

Minor issues:
1. In abstract (line 5) this paper claims that it revisits the difference between KB-based rule induction and LM-based rule induction. However, there is no detailed analysis in the main part of this paper except the statement in Introduction (line 42-44) yet without detailed explanation.

2. In Section 2.1, it presents the general form of a rule which may include multiple atoms in the body. However, in Section 2.2 an open rule seems to contain a single atom in its body. Therefore, it seems that the proposed method can not induce such a rule with multiple atoms in its body in the experiments. Am I correct?

3. In line 106, given an ins, r_p and r_h are not neccessary to be independent. In this sense, the Eq.(3) ideally does not always hold but holds on basis of an independent assumption.


**Main Review:**

One of my main concern is that the baseline Prompt is too weak.
1. According to ablation study in Table 1, the main gains are contributed to the continue training P(r|ins) and P(ins|r) on top of extra corpora (Wikipedia and Bookcorpus). However, the baseline prompt is not conducted with continue training at all.

2. There is a clear gap between the inference and training for the baseline prompt.
During the training, LM is trained on natural sentences, but only some of them satisfy the pattern "if ... then ...". In addition, since r_p is not a natrual sentence, a prompt "if r_p then <mask>" is not a natural sentence during inference. As a result, such a gap in baseline leads to weak performance.

Therefore, to make the baseline prompt more plausible, firstly, it should collect a set of such sentences "if sent1 then sent2" and remove entities from sent1 and sent2. Then it continues to train LM on top of this dataset.




**Time Spent Reviewing:**

12

---

> ### Author Response · Authors · 2021-08-10
> **Response to Reviewer cSoa**
>
> We thank the reviewer for the valuable feedback. Our responses to the questions of the reviewer are as below:
>
> **Main Concern: To make the baseline prompt more plausible, firstly, it should collect a set of such sentences "if sent1 then sent2" and remove entities from sent1 and sent2. Then it continues to train LM on top of this dataset.**
>
> **Response:** Thanks for providing a stronger baseline! We have fine-tuned the prompt baseline as you suggested. The results are shown below, **"new" represents the results of Prompt after continuing pre-training**. This new baseline still performs worse than Orion. We consider this is because it requires the training samples to contain the full if-then rules, which are uncommon in the pre-training corpus. In contrast, our method only requires the sentence to include either the body atom or the head atom, which substantially lowers the requirement for the pre-training corpus.
>
> ```
> ----------------------------------------------------------------------------
>   OpenRule155    BLEU-1   BLEU-2   BLEU-4   ROUGE-L   METEOR   self-BlEU-2
>   Prompt         17.77    3.65     0.48     18.65     12.94      86.63
>   Prompt (new)   20.95    7.58     0.86     22.37     17.24      82.13
>   Orion          45.41    21.29    1.30     50.37     40.41      90.94
> ----------------------------------------------------------------------------
> ```
>
> **Minor Issue 1: In abstract (line 5) this paper claims that it revisits the difference between KB-based rule induction and LM-based rule induction. However, there is no detailed analysis.**
>
> **Response:** This claim may have been misunderstood by the reviewer. We would like to clarify that, by revisiting the difference between KB-based and LM-based methods, what we want to conclude and highlight is that the current LM-based methods limit from "learning rules from rules" (line 7-10). Namely that the major difference between LM-based and KB-based methods is, current LM-based methods learn rules from annotated rules, while the traditional KB-based methods induct rules from data. We believe the latter one is more reasonable and should be applied in the LM-based methods. These are elaborated in detail in line 40-53.
>
> **Minor Issue 2: It seems that the proposed method cannot induce such a rule with multiple atoms in its body.**
>
> **Response:** Our rules could be extended to the case of multiple body atoms, although this is not the focus of this paper. Please refer to the general response for our solution and the problems that might be encountered.
>
> **Minor Issue 3: In line 106, given an ins, $r_p$ and $r_h$ are not necessary to be independent.**
>
> **Response:** As we explain in line 105-106, given the instantiation, then their relation is known. So $r_h$ only depends on $ins$. For example, given $ins$=(Steve Jobs, Apple), their relationship is deterministic. Therefore, the assumption is reasonable.

---

> > ### Comment · Reviewer_cSoa · 2021-08-12
> > **About the new experiments**
> >
> > It is nice to see the new experiments. However, I may not increase the overall scores. In my opinion, the critical points for the proposed method is continue training with extra-training data.  In addition, I read the negative comments from other reviewers and I agree with them.

---

> > > ### Author Response · Authors · 2021-08-25
> > > **Response about the continue training**
> > >
> > >
> > > We thought that the reviewer may consider the continue training critical based on Table 1. We would like to clarify that **the continue training plays the role of denoising the generative process in Eq. (3)** as we illustrated in line136-142. Thus, it is not the focus of our proposed Orion method. The critical point of our method is to discover commonalities from LMs through unsupervised instantiation and generation, instead of learning rules from rules as in the previous LM-based rule generation.
> > >
> > > Indeed, Table 1 indicates that continue training is necessary for Orion to improve the effect. However, this is actually because the generative results of vanilla LM (i.e., BART) are noisy for $P(ins|r_p)$ and $P(r_h|instance)$ in Eq. (3). It often generates undesirable texts conflicting with Eq. (3). To give reviewers more direct insight, we compare the predictions for $P(r_h|instance)$ by the vanilla LM and the LM after continuing training as follows:
> > >
> > > ```
> > > +--------------------------------------------------------------------------------+
> > > |               Top 3 Generation of P(r_h|instance = (Berlin, Germany))          |
> > > +------------------------------------+-------------------------------------------+
> > > | Vanilla BART:                      | continue pre-training BART:               |
> > > +------------------------------------+-------------------------------------------+
> > > |                                    | 1. Berlin is the largest city in Germany. |
> > > | 1. Ber in Germany.Berlin (...)...  |                                           |
> > > |                                    | 2. Berlin was a major center of the       |
> > > | 2. BerGesundheit, Germany, Germany |    printing industry in Germany.          |
> > > |                                    |                                           |
> > > | 3. BerBerlin - Germany.com.au      | 3. Berlin has one of the highest          |
> > > |                                    |    rates of unemployment in Germany.      |
> > > +------------------------------------+-------------------------------------------+
> > > ```
> > >
> > > The vanilla LM is often unable to generate relation descriptions of two entities, namely that it is noisy for $P(r_h|instance)$. In contrast, continue training reduces the noise by ensuring the generated text describe the relation of given instances. Therefore, the continue training is conducted for denoising.
> > >
> > > Besides, the extra training data has no “magic”. It is only constructed by using Spacy on the public corpus (Wikipedia and BookCorpus) to conduct NER tagging. For example, to train the LM for $P(r_h|instance)$, the sentence *“Steve Jobs is the founder of Apple”* will be converted into *“Steve Jobs <mask> Apple”* via NER for training. The continue training corpus itself does not contain any annotated rules.
> > >
> > > We hope that this will help you better understand the role of continue training for our method. And we are willing to discuss with you if you have further confusion about our work.

---

### Official Review · Reviewer_Vt5F · 2021-07-19

**Rating:** 7
**Confidence:** 3

**Summary:**

The recent use of massive language models to solve tasks normally associated with knowledge bases and structured reasoning has opened up inquiry into what kinds of knowledge precisely are stored in LMs. This paper examines the problem of inducing natural-language rules from pre-trained LMs, which is of great interest as large LMs make their way into a variety of products and applications. They argue against current methods for generating rules from KMs and propose a new method called Orion which performs well in downstream tasks. This method uses the masked language modeling objective, which LMs already use for training, to calculate probabilities P(hypothesis|premise).

**Limitations And Societal Impact:**

As language models are only as good as the language they are trained on, rules extracted from them could accordingly be biased or unethical. However, a method for explicitly extracting and diagnosing these problems seems far superior to allowing them to persist latently in opaque model parameters.

**Main Review:**

Updated review and lowered score from 8 to 7 to take into account some of the weakness in evaluation (comparing to models with less training data) pointed out by other reviewers.

**Originality**

The approach to natural language rules by way of decoding the premise/hypothesis probabilities into applicability and instantiation, and analogies to support in KB-based rule induction are compelling. The overall idea of creating a universal rule set in natural language is ambitious and original.

**Quality**

The paper presents a reasonable search-based strategy for evaluating the probabilities and finding likely rules. As beam search has shown some diversity issues with language models, it might have been nice to see approaches based on e.g. nucleus sampling.  They do extensive evaluation, although in somewhat nonstandard settings as might be expected with a new idea such as this.

**Clarity**

Minor grammar/spelling issues. Some typos throughout, such as “Geedily (sic)” in algorithm 1, some switching between past and present tenses in section 3, etc.

**Significance**

Large language models are being increasingly used in industrial applications and to provide state of the art results in pertaining natural language tasks. Both understanding the knowledge contained in these models, as well as extracting that knowledge in useable forms, are of significant interest to the community.

**Time Spent Reviewing:**

1

---

> ### Author Response · Authors · 2021-08-10
> **Response to Reviewer Vt5F**
>
> We thank the reviewer for the positive review and constructive advice.
>
> **Q1. It might have been nice to see approaches based on e.g. nucleus sampling.**
>
> **Response:** Thanks for your comment! Replacing beam search with nucleus sampling is a straightforward while interesting trial. We have added experiments to use nucleus sampling in Orion. The results are shown in the table below. Nucleus sampling and beam search perform comparably.
> ```
> ----------------------------------------------------------------------------------------
>   OpenRule155                BLEU-1   BLEU-2   BLEU-4   ROUGE-L   METEOR   self-BlEU-2
>   Orion (nucleus sampling)   45.68    20.94    1.20     50.50     39.76    90.95
>   Orion                      45.41    21.29    1.30     50.37     40.41    90.94
> ----------------------------------------------------------------------------------------
> ```

---

### Author Response · Authors · 2021-08-10
**General Response**

### **How to Generalize Rules with Multiple Body Atoms and More Variables**

In this paper, we study rules with one body atom and two variables. If we could extend the existing rules to multiple body atoms and more variables, their expressiveness will be enhanced. In line 133-135, we discuss how to introduce more variables. Now we further discussed how to generate rules for multiple body atoms, and the problems that might be encountered.

Consider the rule:

($x$, is an advisor of, $y$) $\land$ ($x$, graduates from, $z$) $\rightarrow$ (y works at z).

Given its body, a straightforward idea to generate the head is to construct two separate probabilistic prompts: “$mask_x$ is an advisor of $mask_y$” and “$mask_x$ graduates from $mask_z$”, and then conduct logical $\land$ operation on them. However, the probability distributions of prompts are opaque, which makes the $\land$ operation intractable.

To represent the $\land$ operation, our idea is to use the power of the natural language to avoid the logical $\land$. We construct a single prompt:
“$mask_x$ is advisor of $mask_y$ and $mask_x$ graduate from $mask_z$.”
The semantics of this prompt is equivalent to the above logic $\land$ expression. Since it is a single prompt, we could still apply the decoding method proposed in this paper to generate the head: $x$ works at $z$.

Notice that the above method introduces a new problem, namely how to ensure the prompt has the same instantiation for two positions of $mask_x$. This could be achieved by adding a constraint to the decoding during instantiation. Besides, as the complexity of the prompt increases, whether the pre-training corpus suffers from sparsity problems will also need to be empirically verified. We leave this as future work.

---

### Decision · Program_Chairs · 2021-09-28

**Decision:**

Accept (Poster)

**Comment:**

This paper introduces a method for extracting and using rules for relation extraction using a pretrained masked language model. This paper was divisive, and the ultimate decision was borderline: Reviewers generally agreed that this line of research is novel and potentially valuable, but were split on the degree to which the experiments back up the main claims of the paper. I'd encourage the authors to revise their paper for clarity, with a focus on the more unusual design choices.

**Consistency Experiment:**

NeurIPS has a long history of experimentation. In 2014, NeurIPS ran an experiment in which 10% of submissions were reviewed by two independent committees to quantify the randomness in the review process. This year, we repeated a variant of this experiment to see how the quality of the review process has changed over time.  This paper was part of the experiment and was therefore assigned to two committees (consisting of reviewers, an Area Chair, and a Senior Area Chair) that reached independent decisions.  If both committees made the same recommendation, this recommendation was followed. If a single committee recommended acceptance, the paper was accepted (with the exception of a few cases in which the other committee identified what we considered a fatal flaw, e.g., an error in a key result).

This copy’s committee reached the following decision: **Reject**

The other committee assigned to the paper recommended **Accept (Poster)**.  You can find the other set of reviews, along with any follow up discussion with the authors here:
https://openreview.net/forum?id=Tku-9lhJC5